# Rapid Simultaneous Detection of the Clinically Relevant Carbapenemase Resistance Genes *bla*KPC, *bla*OXA48, *bla*VIM and *bla*NDM with the Newly Developed Ready-to-Use qPCR CarbaScan LyoBead

**DOI:** 10.3390/ijms26031218

**Published:** 2025-01-30

**Authors:** Martin Reinicke, Celia Diezel, Salma Teimoori, Bernd Haase, Stefan Monecke, Ralf Ehricht, Sascha D. Braun

**Affiliations:** 1Leibniz Institute of Photonic Technology, Leibniz Centre for Photonics in Infection Research (LPI), 07745 Jena, Germany; martin.reinicke@leibniz-ipht.de (M.R.); celia.diezel@leibniz-ipht.de (C.D.); stefan.monecke@leibniz-ipht.de (S.M.); sascha.braun@leibniz-ipht.de (S.D.B.); 2InfectoGnostics Research Campus, 07743 Jena, Germany; 3biotechrabbit GmbH, 12489 Berlin, Germany; salma.teimoori@biotechrabbit.com (S.T.); bernd.haase@biotechrabbit.com (B.H.); 4Institute of Physical Chemistry, Friedrich-Schiller University, 07743 Jena, Germany

**Keywords:** carbapenemase detection, LyoBead multiplex assay, carbapenemase resistance, multi-drug resistant gram-negative bacteria, ready-to-use, point-of-care diagnostics

## Abstract

Antibiotic resistance, in particular the dissemination of carbapenemase-producing organisms, poses a significant threat to global healthcare. This study introduces the qPCR CarbaScan LyoBead assay, a robust, accurate, and efficient tool for detecting key carbapenemase genes, including *bla*KPC, *bla*NDM, *bla*OXA-48, and *bla*VIM. The assay utilizes lyophilized beads, a technological advancement that enhances stability, simplifies handling, and eliminates the need for refrigeration. This feature renders it particularly well-suited for point-of-care diagnostics and resource-limited settings. The assay’s capacity to detect carbapenemase genes directly from bacterial colonies without the need for extensive sample preparation has been demonstrated to streamline workflows and enable rapid diagnostic results. The assay demonstrated 100% specificity and sensitivity across a diverse range of bacterial strains, including multiple allelic variants of target genes, facilitating precise identification of resistance mechanisms. Bacterial strains of the species *Acinetobacter baumannii*, *Citrobacter freundii*, *Escherichia coli*, *Enterobacter cloacae*, *Klebsiella pneumoniae* and *Pseudomonas aeruginosa* were utilized as reference material for assay development (n = 9) and validation (n = 28). It is notable that the assay’s long shelf life and minimal operational complexity further enhance its utility for large-scale implementation in healthcare, food safety, and environmental monitoring. The findings emphasize the necessity of continuous surveillance and the implementation of rapid diagnostic methods for the effective detection of resistance genes. Furthermore, the assay’s potential applications in other fields, such as toxin-antitoxin system research and monitoring of resistant bacteria in the community, highlight its versatility. In conclusion, the qPCR CarbaScan LyoBead assay is a valuable tool that can contribute to the urgent need to combat antibiotic resistance and improve global public health outcomes.

## 1. Introduction

The global emergence and spread of antibiotic resistance (AR) in multidrug-resistant (MDR) bacteria represents a significant threat to public health, undermining the effectiveness of antibiotics and resulting in increased morbidity and mortality [1,2]. Gram-negative bacteria, including *Escherichia coli*, *Klebsiella pneumoniae*, *Acinetobacter baumannii*, and *Pseudomonas aeruginosa*, are especially concerning due to their relevance in causing various infections ranging from uncomplicated UTI to life-threatening sepsis [3,4]. These bacteria frequently exhibit resistance to multiple antibiotic classes, including last-resort carbapenems. The latter can be caused by a production of carbapenemase enzymes encoded by genes, such as *bla*KPC, *bla*NDM, *bla*VIM, and *bla*OXA-48. These enzymes enable the bacteria to hydrolyze carbapenems and other β-lactams, further complicating treatment and increasing healthcare costs, length of hospital stays, and the risk of complications due to treatment failures. Prior studies indicate that *bla*KPC, *bla*NDM, *bla*VIM and *bla*OXA-48 are among the most prevalent carbapenemase genes globally, exhibiting regional variations in prevalence that are influenced by local epidemiological factors. Their prevalence varies by region and is influenced by local epidemiological factors. Continuous surveillance and rapid diagnostic methods targeting these specific genes are crucial for effective management and control of multidrug-resistant bacterial infections worldwide [5,6,7,8].

In addition, the mobility of these resistance genes, often located on mobile genetic elements (MGEs), further complicates the situation by facilitating their rapid spread among bacterial populations and across species barriers. MGEs such as plasmids, transposons, and integrons play a significant role in the dissemination of carbapenemase genes as they enable horizontal gene transfer and allow bacteria to acquire resistance genes from one another regardless of species barriers [9]. This characteristic is particularly concerning for public health because it can lead to the emergence of new resistant strains in various environments, including hospitals and communities. Many carbapenemase genes are encoded on plasmids, which are circular DNA molecules that can replicate independently within bacterial cells. Plasmids facilitate the horizontal transfer of resistance genes between different bacterial species. Transposons are segments of DNA that can move within and between genomes. For instance, Tn4401 is commonly associated with *bla*KPC and is found in various plasmid incompatibility groups, whereas Tn125 is linked to *bla*NDM, often accompanied by insertion sequences like ISAba125 that enhance its mobility [10,11,12]. Carbapenemase genes are found in a variety of gram-negative bacteria and the co-occurrence of multiple carbapenemase genes within a single bacterial isolate is increasingly common. The interaction between carbapenemase genes and mobile genetic elements significantly exacerbates the problem of antibiotic resistance. The ability of these genes to spread rapidly across different bacterial species through MGEs not only complicates treatment strategies but also poses a serious threat to public health globally [13,14,15]. Understanding the mechanisms behind this dissemination is crucial for developing effective control measures against carbapenem-resistant infections.

Contributing to this global crisis are factors such as the overuse of antibiotics in healthcare and agriculture, where antibiotics are widely applied as growth promoters and for disease prevention, which selects for resistant bacteria that can be transmitted to humans via the food chain [16,17]. International migration [18] and war [19,20] further exacerbate the situation, facilitating the rapid spread of resistant bacteria and introducing new resistance mechanisms across borders and regions. These complex factors make the containment of MDR bacteria an urgent global priority [21,22].

The necessity of rapid detection and characterization of carbapenemase genes is critical for infection control, timely treatment, and epidemiological surveillance. Traditional methods for detecting carbapenemase-producing Enterobacteriaceae (CPE), including culturing and susceptibility tests (e.g., VITEK2 or e-Tests), are reliable but often time-consuming and labor-intensive, which can delay appropriate interventions [23,24]. In contrast, molecular techniques, such as multiplex real-time PCR, offer a faster and more specific approach for the simultaneous detection of multiple resistance genes [25,26]. These molecular methods significantly support responsive infection management and control, especially in intensive care settings where time is critical [26].

Building on previous work by Weiss et al. [27], which developed a multiplex real-time PCR assay targeting *bla*KPC, *bla*NDM, *bla*VIM, and *bla*OXA-48, this study introduces an enhanced version utilizing lyophilized qPCR beads. The objective of this project was to develop a robust test from the extensively tested multiplex qPCR with minimal laboratory effort. The emphasis was placed on straightforward handling, economic usability in routine laboratories and high detection accuracy. This upgrade and adaptation allow point-of-care testing (POCT) in different formats (as different thermocycling devices or even POC-cartridges), as these stable lyophilized kits contain all necessary reagents and can be directly applied to bacterial cultures, simplifying workflows and improving ease of use.

## 2. Results

### 2.1. Evaluation of qPCR Master Mixes

The performance of all four ready-to-use qPCR master mixes, 2X Hot-Start PCR Master Mix, 2X YourTaq™ Hot-Start PCR Master Mix, 4X Capital™ qPCR Probe Master Mix and 2X 5Star™ qPCR Probe Master Mix, were tested by preparation of target calibration curves with standard 10-fold dilutions of genomic DNA from bacterial strains harboring the relevant carbapenemase resistance genes. The 2X 5Star™ qPCR Probe Master Mix and 4X Capital™ qPCR Probe Master Mix produced the highest accuracy with a correlation coefficient higher than 0.99 for all four target genes in the calibration curves tested in a multiplex mono template assay (Figure 1).

No false positive signals were detected in the multiplex assay, which showed the high specificity of the different primer probe sets to the regarding target genes. To evaluate the performance and robustness of the multiplex assay, multi-target experiments were performed with genomic DNA from different full genome sequenced reference strains. The template consisted of a genomic DNA mixture which contained all four carbapenemase genes as template. The comparison with the single template experiments resulted in only slight differences in the form of a minimal slope increase of the calibration curves for each target gene and showed the ability of a robust and sensitive multi-target detection in one sample (Figure 2).

### 2.2. Lyophilized qPCR and Complete qPCR LyoBead CarbaScan Var I and II

Comparative experiments with liquid and lyophilized variants of the predefined master mixes with and without primer and probes have shown that the lyophilization process had only minor influence on the total performance of the qPCR-based detection of the resistance genes. Further, the variants of the complete qPCR LyoBeads are named qPCR CarbaScan LyoBead Var I for the 5Star™ qPCR Probe Master Mix based and Var II for the Capital™ qPCR Probe Master Mix based beads. All qPCR mix variants were supplemented with the same template of genomic DNA with a concentration of 100 GE/µL and a template volume of 2 µL per reaction (Figure 3).

For evaluation of the qPCR CarbaScan LyoBead variants, whole genome sequenced reference strains were used as template to perform colony qPCR without any sample preparation step before. In sum, 30 different reference strains were tested to evaluate the detection performance of both qPCR CarbaScan LyoBead variants. For both formulations of the qPCR CarbaScan LyoBead, a specificity of 98.9 up to 100%, sensitivity of 97 up to 100%, and accuracy of 98.3 up to 100% were reached, which reflects a high level of accordance between the sequencing and qPCR assay results (Table 1). With the qPCR CarbaScan LyoBead Var II, all target genes of the reference strains were detected without any false positives. There was no detectable cross-hybridization in the samples used. In all strains carrying more than one resistance, these resistances were correctly identified.

Analysis with the qPCR CarbaScan LyoBead Var I resulted also in a high rate of concordance to the nanopore-based resistance prediction, except for a false positive finding of *bla*NDM and false negative of *bla*OXA-48 as allelic variant *bla*OXA-232, both in *Klebsiella pneumoniae* strains (ID 95573 and 280220). In other *Klebsiella pneumoniae* strains with the same allelic variant of *bla*OXA-48, the resistance genes were correctly identified. The simultaneous occurrence of an overestimation and a lack of detection led to a reduction in the specificity and sensitivity of this assay variant.

## 3. Discussion

Detection of carbapenemase genes is of critical clinical importance due to their strong association with antibiotic resistance not only to carbapenems but also to other beta-lactams, a major challenge in the treatment of infections caused by multidrug-resistant organisms (MDROs). Carbapenemase-producing bacteria are able to resist last-line antibiotics, making infections difficult to treat and increasing the risk of poor patient outcomes [28].

Rapid detection of these genes directly from suspect bacterial colonies, without the need for extensive sample preparation, offers both economic and practical advantages. This streamlined approach enables faster diagnostic results, which are critical for making timely treatment decisions and implementing effective infection control measures [29].

Carbapenemase genes are often carried on mobile genetic elements (MGEs) that facilitate their transfer between bacterial species and the spread of resistance [30,31]. The ability to detect these genes without prior knowledge of the bacterial species broadens the applicability of diagnostic assays and highlights the need for robust detection methods.

In this study, we developed an easily feasible and fast qPCR CarbaScan LyoBead assay, based on predefined qPCR master mixes, and demonstrated its high robustness and accuracy in detecting clinically relevant carbapenemase resistance genes.

Cultivation-based assays, including the Carba-NP test [32], the Hodge test [33], the carbapenem inactivation method [34] and the agar diffusion assay [35], provide phenotypic evidence of carbapenem resistance. However, these assays require an incubation period that is longer than that of molecular-based methods. Furthermore, these assays do not provide any information about the resistance genes present, which may be relevant for therapy with regard to the selection of antibiotics [36,37]. Assays based on qPCR can provide this information about the type of resistance genes and thus provide therapy-decisive facts, and they are comparably accurate as the cultivation-dependent methods [38]. Numerous established qPCR assays for clinical samples [25] or environmental monitoring [39] are based on liquid qPCR master mixes, which must be transported and stored under refrigeration. In the case of monoclonal pure cultures, the DNA must be prepared, which can be more or less time-consuming. The developed qPCR CarbaScan LyoBead facilitates direct screening with cell material from the bacterial culture. The qPCR CarbaScan LyoBead Var II demonstrated 100 percent specificity and sensitivity in identifying carbapenemase resistance genes, including *bla*KPC, *bla*NDM, *bla*OXA-48, and *bla*VIM, across a range of gram-negative bacterial strains. Furthermore, the assay demonstrated high specificity in the detection of various allelic variants of these resistance genes. For example, variants 1, 2, 4, and 6 were successfully detected for *bla*VIM; variants 48, 181, 232, and 163 for *bla*OXA-48; variants 2 and 3 for *bla*KPC; and variants 1, 2, 4, 5, and 7 for *bla*NDM.

The false negative finding of *bla*OXA-232 in Klebsiella pneumoniae ID 280220 with the qPCR CarbaScan LyoBead Var I (5Star based) is most likely due to a too low fluorescence intensity of the probe in combination with this strain. The beginning of the rise of the fluorescence curve corresponds to that of other *bla*OXA-232 in the evaluation, but the lift was below the threshold. The false positive signal of *bla*NDM in *Klebsiella pneumoniae* ID 95573 was reproducible and the cause is unknown. The bioinformatic sequence analysis showed no potential binding sites for the primer probe set of *bla*NDM in the DNA of this bacterial strain.

The LyoBead assay is distinguished by several advantageous characteristics. The assay is simple to operate, does not necessitate refrigeration for storage, is stable for a long time, and can be used in conjunction with commercially available qPCR (quantitative PCR) cyclers that have five detection channels. The formulation as a lyophilized bead allows for storage times of at least 24 months at ambient temperature. For comparable products, the manufacturer provides a warranty of shelf lives ranging from 36 to 48 months, provided that the product is adequately packaged and sealed. The assay has been demonstrated to reduce the time and effort required for sample preparation, thereby facilitating a more efficient analysis of common carbapenemase resistances in colony PCR format. The assay’s simplicity and reduced number of work steps contribute to a reduction in the likelihood of errors, such as pipetting errors or contamination. However, when using the qPCR CarbaScan LyoBead assay in a colony PCR format, it is essential to employ fresh bacterial cultures to avoid loss of specificity and sensitivity; using older culture material for the evaluation of the CarbaScan assay has been shown to result in reduced sensitivity and possible false negatives (Appendix A Table A1).

The simplicity, long shelf life and good performance of the qPCR CarbaScan LyoBead assay make it an excellent starting point for the development of a standardized assay for the rapid detection of resistance genes. The stability of these lyophilized kits also optimizes transport and storage, making them accessible and practical for diverse healthcare environments, particularly in resource-limited settings. Compared to conventional diagnostic methods, this enhanced assay offers a fast, reliable, and cost-effective solution for detecting carbapenemase-producing organisms at the point of care.

By targeting the four clinically significant carbapenemase genes in a single, rapid assay, this advanced method supports healthcare providers in managing and controlling MDR bacterial infections on a global scale. The lyophilized format of the upgraded assay enhances convenience and efficiency, thereby supporting improved antibiotic stewardship, effective infection control measures, and contributing to global public health efforts against the spread of MDR bacteria.

Although the qPCR CarbaScan LyoBead Var II assay provides an accurate and efficient method for detecting carbapenemase resistance genes, the broader implications of their persistence and dissemination, particularly in conjunction with toxin-antitoxin systems, highlight the necessity for comprehensive strategies to combat antibiotic resistance.

The co-existence of carbapenemase genes and toxin-antitoxin (TA) systems on a single plasmid presents a considerable challenge for the management of antibiotic resistance. TA systems permit bacteria to retain plasmids carrying resistance genes even in the absence of antibiotic selection pressure, thereby representing a significant public health risk [40,41]. This mechanism confers a selective advantage upon plasmid-bearing bacteria, thereby ensuring their survival even in environments without antibiotics. The capacity of community-associated bacterial strains to retain these plasmids enhances the likelihood of the dissemination of resistance genes within healthy populations. The dissemination of multidrug-resistant organisms (MDROs) is further facilitated by horizontal gene transfer, which allows these organisms to infect otherwise healthy individuals and lead to outbreaks that are difficult to control. For example, non-clinical strains of bacteria that harbor carbapenemase genes have been identified in food sources and environmental samples, which serves to illustrate the broader dissemination risks. These findings emphasize the necessity for the implementation of efficacious surveillance and intervention strategies to prevent the establishment of resistance genes within community populations [42,43].

The objective of this study was to develop a universal qPCR-based test that can be utilized across a range of thermocyclers. However, due to technological constraints, the number of target genes incorporated into the test was limited by the number of detection channels available for qPCR. Consequently, the four most prevalent carbapenemase classes with global significance were selected for analysis [7]. However, the global distribution of carbapenemase classes exhibits significant disparities [44,45]. For instance, *bla*IMP and *bla*GES, which were not included in the panel, are more prevalent in Asia [25]. To ensure the monitoring of these regionally significant carbapenemases, it would be necessary to extend the panel, potentially through the incorporation of an additional LyoBead targeting these classes, as demonstrated in the study of Yoshioka et al. (2021) [25]. While the panel is designed to detect the most prevalent and known allelic variants, the possibility of undetected variants due to sequence disparities, such as single-nucleotide polymorphisms (SNPs), cannot be ruled out. Additionally, the potential emergence of novel allelic variants resulting from ongoing mutations could extend beyond the scope of the panel’s detection. This, in turn, could lead to diminished assay performance if SNPs arise in the primer and probe binding sites, underscoring the necessity for continuous assay optimization and post-market surveillance.

The assay demonstrated high effectiveness as a culture confirmation test for carbapenem-resistant Gram-negative bacterial strains harboring *bla*OXA-48, *bla*VIM, *bla*NDM, and *bla*KPC. The assay’s performance was further validated by the reliable detection of all target markers in the reference strains, underscoring its reliability and precision. Specifically, the CarbaScan LyoBead Var II exhibited exceptional specificity, sensitivity, and accuracy, thereby minimizing the likelihood of false-positive or false-negative results. However, the potential for false-positive results, if rare, could lead to the unnecessary administration of antibiotics and overtreatment. Conversely, false-negative results pose a greater concern, as they could delay the initiation of appropriate therapy and adversely impact clinical outcomes. The application of native clinical samples, such as nasopharyngeal or rectal swabs, sputum, or urine, as templates remains unexplored and would require additional experiments to evaluate its feasibility and reliability. To implement the assay in routine clinical diagnostics, further large-scale studies would also be necessary to validate its performance and support potential regulatory approval.

The assay discussed herein is optimal for efficient, cross-species screening of bacterial pathogens in a multitude of settings, including hospitals, animal husbandry, and food monitoring. The assay is cost-effective to implement due to its straightforward handling and potential for large-scale production, making it an appropriate tool for monitoring antibiotic resistance. This makes it a suitable method for the prevention of health-relevant outbreaks by controlling the occurrence and spread of resistance genes. Furthermore, the assay’s capacity to identify resistance genes in a diverse range of bacterial strains establishes it as a valuable diagnostic instrument. This addresses the necessity for continuous surveillance and rapid diagnostic techniques targeting specific carbapenemase genes, which are vital for the management and control of multidrug-resistant bacterial infections on a global scale.

## 4. Materials and Methods

### 4.1. Bioinformatics of Oligonucleotides

We have tested a combination of multiplex qPCR primers and probes for the detection of carbapenemase genes from a previous work for their suitability for use in a LyoBead [27]. As internal process control, *Pseudomonas syringae* (strain DSM10604_170912) was introduced, a widespread bacterial commensal on soybean with no pathogenic association to humans [46].

For the *Pseudomonas syringae* species marker, multi-FASTA alignments were prepared using MAFFT (version v7.475). The primers and TaqMan probe for the *P. syringae* qPCR (Table 2, rows 13–15) were designed using ConsensusPrime (GitHub version 872ec4c) with parameters identical to those of the other primers and probes (e.g., Tm) [47]. TaqMan probes were labelled with different fluorophores covalently attached to the 5′ end. All oligonucleotides were produced by Metabion (Steinkirchen, Germany).

### 4.2. Bacterial Strains and Growth Conditions

To evaluate the method’s effectiveness, various strains from our in-house strain collection were used as reference standards. The evaluation panel included nine reference strains for bead development and an additional thirty strains for validation representing different species: *Acinetobacter baumannii*, *Citrobacter freundii*, *Escherichia coli*, *Klebsiella pneumoniae*, *Pseudomonas aeruginosa* and *Pseudomonas syringae* (Table 3). All strains were cultured on Columbia blood agar (Becton Dickinson, Heidelberg, Germany) overnight at 37 °C (*P. syringae* at 28 °C). Additionally, antibiotic susceptibility testing was performed on each reference strain using the VITEK-2 system, following the manufacturer’s protocol and in accordance with EUCAST guidelines (version 2023). Moreover, all strains underwent next-generation whole-genome sequencing via Oxford Nanopore Technology (ONT) to identify allelic variants and determine whether resistance genes were located on chromosomes or plasmids.

### 4.3. Nucleic Acid Preparation and Nanopore Sequencing

DNA extraction for sequencing with the Oxford Nanopore MinION (Oxford Nanopore Technology, Oxford, UK) was conducted using the Nucleospin Microbial DNA Kit from Macherey Nagel (MN, Düren, Germany). To begin, all bacterial strains were revived from cryo-cultures (Microbank; ThermoFisher Scientific, Waltham, MA, USA) by culturing on blood agar plates at 37 °C overnight. For each strain, a full inoculation loop was resuspended in 500 µL of 1 × PBS (pH 7.4), centrifuged, and the pellet resuspended in 100 µL of buffer BE. The subsequent DNA extraction steps followed the manufacturer’s protocol with two modifications: (1) samples were lysed using a bioshaker (QINSTRUMENTS, Jena, Germany) for 4 min for Gram-negative bacteria at maximum speed, and (2) prior to binding the DNA to the Nucleospin microbial DNA columns, proteinase K was inactivated by incubating the samples at 70 °C for 5 min. After cooling, 4 µL of RNAse (100 mg/mL; Sigma Aldrich, Steinheim, Germany) was added, and samples were incubated at 37 °C for 5 min. The DNA was then eluted twice with 75 µL of nuclease-free water (Carl Roth, Karlsruhe, Germany).

For genome sequencing of the strains, we used MinION flow cells and flongle flow cells (R9.4.1 or R10.4.1). Library preparation was carried out with the 1D Genomic DNA Ligation Kit (SQK-LSK109) along with the Native Barcoding Expansion Kits (EXP-NBD103, EXP-NBD104, and EXP-NBD114). Briefly, size selection and DNA cleanup were performed using Agencourt AMPure XP beads (Beckman Coulter GmbH, Krefeld, Germany) at a 1:1 (v:v) ratio before library preparation. DNA nicks and ends were repaired simultaneously using the NEBNext FFPE DNA Repair Mix and the NEB Next Ultra II End Repair/dA-Tailing Module (New England Biolabs, Ipswich, MA, USA), with an extended incubation time. Barcodes were ligated to the dA-prepared DNA ends, followed by a second AMPure bead clean-up. After a third AMPure purification step, sequencing adapters were ligated to the DNA ends. At the start of sequencing, each flow cell was quality-checked, ensuring at least 1200 active pores. Genomic DNA samples for loading contained approximately 40–60 ng per strain, quantified with the Qubit 4 Fluorometer (ThermoFisher Scientific). Sequencing ran for 72 h, managed by MinKNOW software version 22.05.5. All kits were used according to the manufacturer’s instructions.

The Guppy basecaller (version 4.5.2 + bcc53d392 to 6.0.1 + 652ffd179, Oxford Nanopore Technologies, Oxford, UK) [48] was used to convert MinION raw reads (FAST5) into quality-tagged sequence reads, producing 4000 reads per FASTQ file. Barcode trimming was enabled (models: dna_r9.4.1_450bps_sup.cfg and dna_r10.4.1_e8.2_400bps_sup). Quality-tagged reads were assembled into complete circular contigs for each strain using Flye (version 2.8.3) [49]. Assembly polishing occurred in two steps: initially, four iterative rounds of Racon (v1.4.21) [50] with parameters match 8, mismatch 6, gap 8, and window length of 500, followed by polishing with Medaka (version 1.4.3) [51] using models r941_min_sup_g507 and r10.4.1_e82_400bps_sup_g615. Finally, Abricate (v1.0.0) [52] was used to screen the corrected assemblies for resistance and virulence genes.

### 4.4. Genomic DNA Dilution

To validate and quantify the experimental results, defined 10-fold dilution series of genomic DNA were prepared for each of the nine reference strains. Genomic equivalents (GE) were calculated based on the genome sizes of the sequenced bacterial species and the concentration of prepared genomic DNA samples. Resistance genes were either chromosomally encoded or plasmid-borne, depending on the strain. For resistance markers located on plasmids, this approach enabled a semi-quantitative estimation of plasmid copy numbers in relation to genomic copies.

### 4.5. qPCR Assays for LyoBead Development and Evaluation

The qPCR assays were performed in 20 µL volumes. Four different ready-to-use PCR master mixes commercially available from biotechrabbit GmbH (Berlin, Germany) were used for the assay mix. The following master mixes were employed for the experiments: 2X Hot-Start PCR Master Mix (BR0200205), 2X YourTaq™ Hot-Start PCR Master Mix (BR0202201), 4X Capital ™ qPCR Probe Master Mix (BR0501401) and 2X 5Star™ qPCR Probe Master Mix (BC0502501). The selection of the qPCR master mixes was based on the availability of the mixes in liquid and lyophilized formulation, as well as their performance. Primers were added with a final concentration of 100 nM and probes with 200 nM. The amplification was carried out in QuantStudio5 qPCR cycler (Thermo Fisher Scientific Inc., Waltham, MA, USA). The temperature program started with an initial denaturation at 95 °C for 3 min followed by 40 cycles of 15 s denaturation at 95 °C and annealing and elongation at 61.5 °C for 30 s. To test the performance of mono- and multiplex qPCR, all master mixes were tested in liquid and lyophilized form. The qPCR optimized master mixes 4X Capital ™ qPCR Probe Master Mix and 2X 5Star™ qPCR Probe Master Mix were completed with primer and probes and the final CarbaScan LyoBead were manufactured and supplied by biotechrabbit GmbH, Berlin, Germany. The composition of the various assay formulations (liquid, lyophilized and lyophilized complete) is displayed in Table A2 as an overview.

Mono- and multitarget qPCR assays were performed to validate the individual primer pairs and probes. A calibration curve was prepared for each marker with a 10-fold dilution series from 10^6^ down to 10^1^ GE/µL with a 2 µL template volume of purified genomic DNA from reference strains. To check the specificity of all primer/probe sets, cross-hybridization experiments were performed with all primer/probe sets and all genomic DNA samples from the non-target reference strains (Figure 4). The efficiency of the quantitative polymerase chain reaction was determined based on the slope of the calibration curve.

Evaluation of the complete LyoBead (qPCR CarbaScan LyoBead) was performed by colony PCR using WGS-characterized reference strains (Table 3). Per reaction a complete LyoBead was reconstituted with the corresponding 4× concentrated buffer and 200 GE of genomic DNA from *Pseudomonas syringae* were added as internal process control. The mix was filled up with molecular grade water to a final volume of 20 µL. Template material from a bacterial colony was then transferred with a plastic inoculation needle into the qPCR mix. The samples were processed as described above.

### 4.6. Statistical Analysis

The development of the assay utilized nine bacterial strains characterized by nanopore sequencing, targeting the genes *bla*KPC (n = 2), *bla*NDM (n = 2), *bla*OXA-48 (n = 3), and *bla*VIM (n = 4), with allelic variants *bla*KPC-2, *bla*NDM-1, *bla*OXA-48, and *bla*VIM-1/2. These carbapenemase classes were distributed across *Acinetobacter baumannii*, *Citrobacter freundii*, *Escherichia coli*, and *Klebsiella pneumoniae*. The oligonucleotide panel underwent extensive evaluation during its development using 82 carbapenemase-producing and 50 non-producing bacterial strains, as reported by Weiss et al. (2017) [27]. An additional set of 28 bacterial strains, also characterized by nanopore sequencing, was used for the evaluation of the qPCR CarbaScan LyoBead variants. These strains harbored the target genes *bla*KPC (n = 7), *bla*NDM (n = 6), *bla*OXA-48 (n = 8), and *bla*VIM (n = 8), with an extended range of allelic variants, including *bla*KPC-2/3, *bla*NDM-1/2/4/5/7, *bla*OXA-48/163/181/232, and *bla*VIM-1/2/4/6. The carbapenemase classes in these strains were distributed among *Acinetobacter baumannii*, *Citrobacter freundii*, *Enterobacter cloacae*, *Escherichia coli*, *Klebsiella pneumoniae*, and *Pseudomonas aeruginosa*.

A result comparison was made of the cycle threshold (CT) mean values between the three experimental conditions (liquid, lyophilized, and complete) for each marker. This was accomplished using two-sample independent *t*-tests. The mean values for each condition were then analyzed with a standard deviation and a sample size of three. The statistical analysis included a series of paired comparisons, namely, liquid vs. lyophilized, liquid vs. complete, and lyophilized vs. complete. Statistical significance was determined based on the calculated *p*-values for each comparison. The analysis was conducted using the ttest_ind_from_stats function from the SciPy (v1.8.0, python3.8) statistics library. This methodological approach facilitated the identification of significant differences in CT values across the experimental setups for each marker (Table A3.).

The detection outcomes for each gene of the evaluation experiments were categorized into four groups: True Positives (TP), where a gene was identified by both sequencing and qPCR; True Negatives (TN), where neither method detected the gene; False Positives (FP), where a gene was detected by qPCR but not by sequencing; and False Negatives (FN), where a gene was identified by sequencing but not by qPCR. These classifications were applied to each resistance gene category considered.

Performance metrics were derived for each category based on the TP, TN, FP, and FN values. Sensitivity was calculated as the proportion of True Positives relative to the total number of positives (sensitivity = TP/(TP + FN)), while specificity was determined as the proportion of true negatives relative to the total number of negatives (specificity = TN/(TN + FP)). Concordance (or accuracy) was defined as the proportion of correctly classified genes relative to the total classifications (accuracy = (TP+TN)/(TP + TN + FP + FN)). These metrics were used to evaluate the overall agreement between sequencing and qPCR analysis in detecting antimicrobial resistance genes. The confidence interval (95%) for specificity, sensitivity and accuracy was calculated using the Clopper-Pearson-interval.

## 5. Conclusions

The identification of carbapenemase genes is of fundamental importance in the global management of antibiotic resistance, particularly with regard to the treatment of infections caused by multidrug-resistant organisms (MDROs). The recently developed qPCR CarbaScan LyoBead assay signifies a substantial advancement in this domain. The assay’s simplicity, robustness, and high specificity combine to offer an accurate method for detecting key carbapenemase genes such as *bla*KPC, *bla*NDM, *bla*OXA-48, and *bla*VIM across a wide range of Gram-negative bacterial strains. The assay’s lyophilized bead format ensures long-term stability, eliminates the need for cold-chain storage, and simplifies operational logistics, making it highly suitable for point-of-care applications and resource-limited settings.

However, its scope is limited by the number of detection channels available for qPCR. Expanding the panel with genes for additional carbapenemase classes would enhance its applicability. Future studies should validate its use with native clinical samples and on a larger scale to support regulatory approval and routine clinical implementation.

The assay’s capacity to provide rapid results directly from bacterial colonies, circumventing the necessity for extensive sample preparation, enables a streamlined diagnostic process and facilitates timely therapeutic interventions. This rapid detection capability enhances infection control efforts and aids in reducing the spread of resistance genes. Despite its encouraging performance, the assay is subject to certain limitations, including the occasional occurrence of false negatives and positives, underscoring the necessity for ongoing optimization and validation. Expanding the panel to include additional targets could further enhance its versatility, addressing the diverse global distribution of carbapenemase genes.

Beyond its clinical applications, the assay demonstrates potential for broader applications, including food safety and environmental monitoring, thereby underscoring its versatility in combating the dissemination of resistance genes. The co-existence of carbapenemase genes with toxin-antitoxin systems and their mobility on plasmids further emphasizes the need for robust surveillance strategies to mitigate the public health risks posed by these resistance mechanisms.

In conclusion, the qPCR CarbaScan LyoBead assay offers a valuable diagnostic tool to support global efforts in combating antibiotic resistance. Its capacity to facilitate efficient, large-scale monitoring, in addition to providing a foundation for further assay development, signifies a contribution to enhanced patient outcomes, enhanced infection control measures, and the ongoing fight against the proliferation of MDROs.

## Figures and Tables

**Figure 1 ijms-26-01218-f001:**
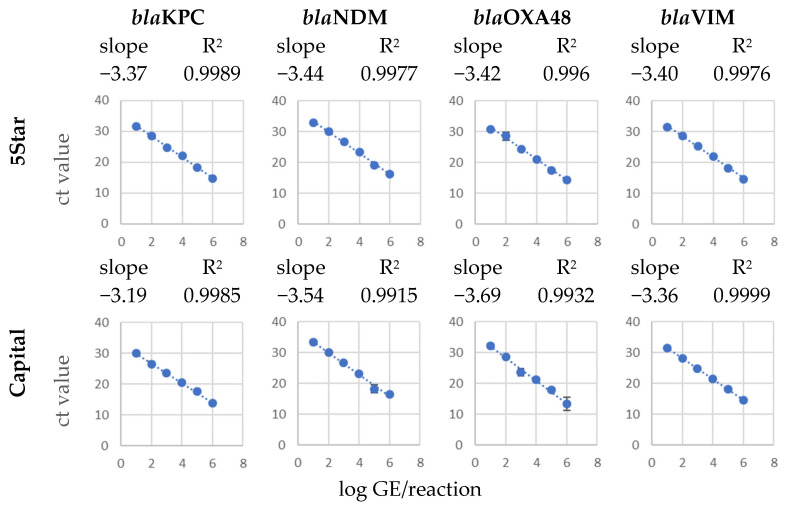
Calibration curves of multiplex qPCR reactions with genomic DNA dilutions (single target) from D2 (2,000,000 GE (genomic equivalents) per reaction) down to D7 (20 GE/reaction) for target genes *bla*KPC, *bla*NDM, *bla*OXA-48 and *bla*VIM with the liquid qPCR master mixes. Slope and regression coefficient indicated a high accuracy of the calibration curves. A 10-fold dilution series of genomic DNA was utilized as a template. The highest concentration employed was 1,000,000 genome equivalents (GE) per microliter, with the lowest concentration being 10 GE per microliter. The assay was performed as two-step qPCR with 2 µL template of purified genomic DNA (from D2 to D7).

**Figure 2 ijms-26-01218-f002:**
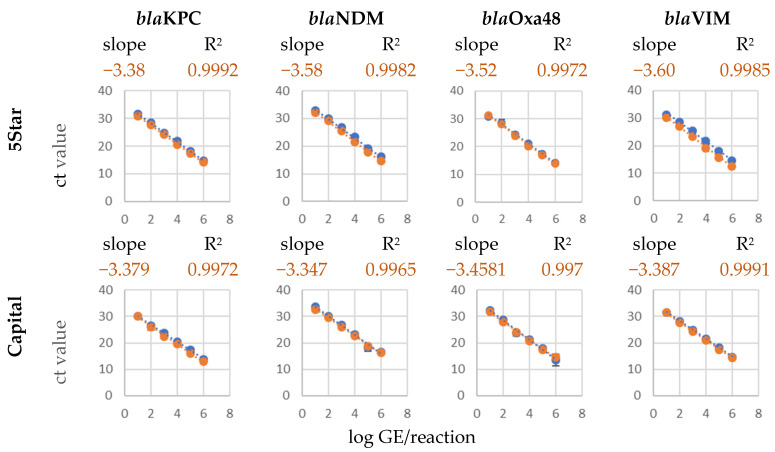
Comparison of the calibration curves of the multiplex qPCR assays with the liquid qPCR master mixes with a single template (only one target gene in the multiplex approach, shown in blue) and with all four target genes in a multi-template approach (shown in orange). Slope and regression coefficient belong to the multi-template curves. The multitarget multiplex quantitative PCR (qPCR) assays were performed in accordance with the same protocol as the single target version.

**Figure 3 ijms-26-01218-f003:**
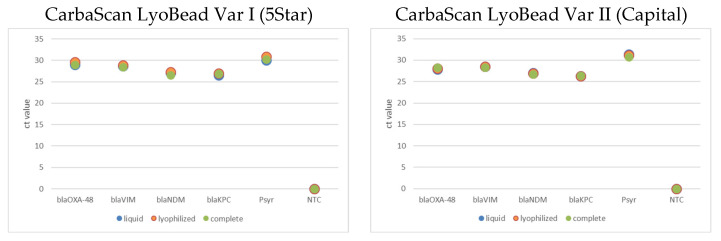
The following figure illustrates the Ct values of the qPCR assays in the formulation’s liquid in blue, LyoBead without primer and probes in orange, and complete LyoBead with primer and probes (Carba Scan LyoBead Var I on the left and II on the right) in green. A mixture of genomic DNA with all target genes at a concentration of 200 GE/reaction was used as multitarget template. IPC (Psyr) was added with the same concentration. The assay was performed as two-step qPCR in accordance with the previous experiments with liquid and lyophilized master mixes.

**Figure 4 ijms-26-01218-f004:**
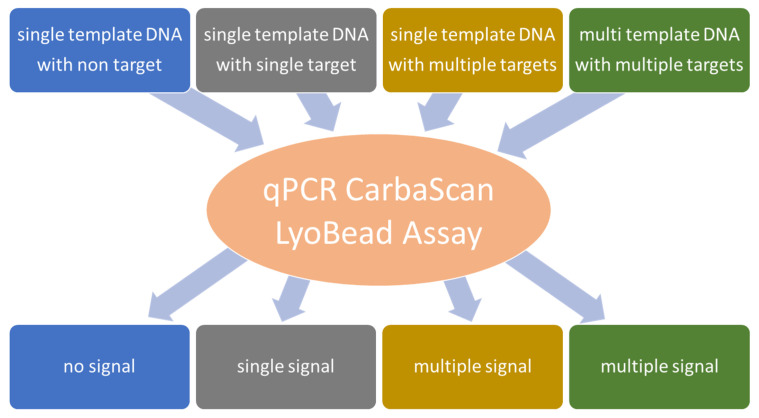
The testing of cross-hybridizations with diverse templates of genomic DNA, incorporating either a singular target gene or multiple target genes within a singular template, is employed to simulate infections with a solitary pathogen. Moreover, a combination of genomic DNA from multiple pathogens, each possessing a distinct target gene, is utilized to emulate an infection involving numerous pathogens.

**Table 1 ijms-26-01218-t001:** Sensitivity, specificity and accuracy of the different versions of the CarbaScan LyoBead in comparison to full genome sequencing.

Version	TP ^1^	FP ^2^	TN ^3^	FN ^4^	Specificity (c. i) ^5^	Sensitivity (c. i.) ^5^	Accuracy (c. i.) ^5^
qPCR CarbaScan LyoBead Var I	32	1	86	1	98.9% (0.938–1.000)	93.9% (0.798–0.993)	97.5% (0.929–0.995)
qPCR CarbaScan LyoBead Var II	33	0	87	0	100.0% (0.958–1.000)	100.0% (0.894–1.000)	100.0% (0.970–1.000)

^1^ TP: True Positive; ^2^ FP: False Positive; ^3^ TN: True Negative; ^4^ FN: False Negative; ^5^ c. i.: confidence interval (α = 0.05).

**Table 2 ijms-26-01218-t002:** List of used oligonucleotides for the multiplex qPCR and formulation of LyoBeads.

#.	Oligo Name	Oligo Sequence	5′ Label	3′ Quencher	T_m_ ^1^	Accession Number
1	blaKPC_fwd	CTGTATCGCCGTCTAGTTCTG			61.9	EU447304.1 [15:896]
2	blaKPC_rev	AGTTTAGCGAATGGTTCCG			62.1	
3	blaKPC_probe	TGTCTTGTCTCTCATGGCCGCTGG	FAM	BHQ-1	75.4	
4	blaNDM_fwd	GCATTAGCCGCTGCATT			63.1	FN396876.1 [2407:3219]
5	blaNDM_rev	GATCGCCAAACCGTTGG			65.7	
6	blaNDM_probe	ACGATTGGCCAGCAAATGGAAACTGG	ROX	BHQ-2	76.1	
7	blaOXA48/181_fwd	TTCCCAATAGCTTGATCGC			63.1	Consensus (OXA-48-group)
8	blaOXA48/181_rev	CCATCCCACTTAAAGACTTGG			62.6	
9	blaOXA48/181_probe	TCGATTTGGGCGTGGTTAAGGATGAAC	HEX	BHQ-1	74.8	
10	blaVIM_fwd	TGGCAACGTACGCATCACC			68.5	Consensus
11	blaVIM_rev	CGCAGCACCGGGATAGAA			67.7	
12	blaVIM_probe	TCTCTAGAAGGACTCTCATCGAGCGGG	Cy5	BHQ-3	73.0	
13	ipc_psyr_fw	GGTTTGGTAGACGGTTCGA			63.0	CP000075.1 [21609082160931]
14	ipc_psyr_rv	GACCGAGAAAGACGTAAGCA			62.4	
15	ipc_psyr_probe	TGGGTCAGGTTGCCCATTGACAGA	TAMRA	BHQ-2	75.5	

^1^ Tm: melting temperature.

**Table 3 ijms-26-01218-t003:** List of bacterial strains used for test development (underlined) and evaluation.

Organism	Strain ID ^1^	Resistance Genes (Location) ^2^
*Acinetobacter baumannii*	95932	
*Acinetobacter baumannii*	215784	*bla*VIM-2 (p)
*Acinetobacter baumannii*	240611	*bla*NDM-1 (c)
*Acinetobacter baumannii*	240737	*bla*NDM-2 (c)
*Acinetobacter baumannii*	301751	*bla*NDM-1 (c)
*Citrobacter freundii*	240619	*bla*OXA-48 (p)*, bla*VIM-1 (p)
*Citrobacter freundii*	242274	*bla*NDM-1 (p)
*Citrobacter freundii*	279615	*bla*VIM-1 (p)
*Enterobacter cloacae*	97966	*bla*OXA-163 (p)
*Escherichia coli*	97947	*bla*NDM-4 (p)
*Escherichia coli*	98219	*bla*NDM-5 (p)
*Escherichia coli*	240608	*bla*OXA-48 (c)
*Escherichia coli*	240615	*bla*KPC-2 (p)
*Escherichia coli*	240776	*bla*NDM-7 (p)
*Escherichia coli*	240780	*bla*VIM-4 (p)
*Escherichia coli*	319495	
*Klebsiella pneumoniae*	79748	*bla*KPC-2 (p), *bla*VIM-1 (p)
*Klebsiella pneumoniae*	95473	*bla*OXA-48 (p)
*Klebsiella pneumoniae*	95506	*bla*OXA-181 (c)
*Klebsiella pneumoniae*	95515	*bla*OXA-232 (p)
*Klebsiella pneumoniae*	95571	*bla*OXA-48 (p)
*Klebsiella pneumoniae*	95573	*bla*OXA-48 (p)
*Klebsiella pneumoniae*	215756	*bla*KPC-2 (p)
*Klebsiella pneumoniae*	219840	*bla*KPC-2 (p), *bla*VIM-1 (p)
*Klebsiella pneumoniae*	223971	*bla*KPC-3 (p)
*Klebsiella pneumoniae*	238631	*bla*KPC-2 (p)
*Klebsiella pneumoniae*	239644	*bla*OXA-48-like (p)
*Klebsiella pneumoniae*	240799	*bla*NDM-1 (p), *bla*OXA-232 (p)
*Klebsiella pneumoniae*	242816	*bla*KPC-2 (p), *bla*VIM-2 (p)
*Klebsiella pneumoniae*	245295	*bla*KPC-3 (p)
*Klebsiella pneumoniae*	272567	*bla*VIM-1 (p)
*Klebsiella pneumoniae*	274401	*bla*KPC-2 (p)
*Klebsiella pneumoniae*	280220	*bla*OXA-232 (p)
*Klebsiella pneumoniae*	280236	
*Pseudomonas aeruginosa*	279584	*bla*VIM-4 (c)
*Pseudomonas aeruginosa*	280207	*bla*VIM-6 (c)
*Pseudomonas aeruginosa*	280228	*bla*VIM-2 (c)
*Pseudomonas syringae* (IPC)	305664	

^1^ Strain ID: identifier internal strain collection; ^2^ (c) chromosomal-encoded, (p) plasmid-encoded.

## Data Availability

The sequences of the reference strain genomes can be accessed under BioProject accession number PRJNA779589.

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
