# Peer review of "Rapid Simultaneous Detection of the Clinically Relevant Carbapenemase Resistance Genes blaKPC, blaOXA48, blaVIM and blaNDM with the Newly Developed Ready-to-Use qPCR CarbaScan LyoBead"

_ijms, 2025, doi:10.3390/ijms26031218_

Round 1

Reviewer 1 Report

Comments and Suggestions for Authors

Reinicke et al. have aimed to develop a rapid method for simultaneous detection of clinically relevant carbapenemase resistance genes using the newly developed ready-to-use qPCR CarbaScan LyoBead. This is an interesting piece of work performed methodically and well written. However, the following issues need to be elaborated for a possible publication in IJMS:

1.       In the title, the “parallel” can be replaced with “simultaneous”.

2.       In the abstract, it is important to include the exact sample size and bacterial strains tested.

3.       In the keywords, “point-of-care diagnostics” should be added.

4.       In the introduction, a formal objective statement should be added at the end.

5.       In the methods, explain the reasons for choosing specific master mixes and provide the details of cross-hybridization experiments with a flow diagram.

6.       In the methods, the authors should ensure adding at least one reference for each reported protocol.

7.       In the methods, a statistical analysis section should be added to provide the details of number of replicates and significance of data.

8.       In Figures 1-3, the experimental conditions should be briefly added in the respective captions.

9.       In Table 1, a column showing the confidence interval should be included.

10.    In Table A1, it is important to highlight the significance of culture age affecting assay results.

11.    In all Tables and Figures, the abbreviations used should be provided in full form in the respective table footnotes and figure captions.

12.    In the results, some more explanation on the implications of false positives and negatives for specific strains should be provided.

13.    In the discussion and conclusion, the limitations of this assay and future perspectives on application of this assay should be included.

Author Response

Dear Reviewer,

We would like to sincerely thank you for taking the time to review our manuscript and for providing detailed and constructive comments. We truly appreciate your thoughtful feedback and the valuable insights you have shared to improve the quality of our work.

We have carefully addressed each of your comments and incorporated the necessary changes into the manuscript. For your convenience, we have attached a detailed point-by-point response to all the comments, along with the revised manuscript.

We hope that the revisions meet your expectations, and we are happy to address any additional questions or suggestions you may have.

Thank you again for your time and effort.

Best regards,

Martin Reinicke

Reviewer 2 Report

Comments and Suggestions for Authors

The manuscript entitled “Rapid parallel detection of the clinically relevant car- 2 bapenemase resistance genes blaKPC, blaOXA48, blaVIM and 3 blaNDM with the new developed ready-to-use qPCR 4 CarbaScan LyoBead” describes a robust assay for detecting key carbapenemase genes.

I have only a minor revision regarding the validation study. Could you please specify in metarial and methods how you calculated % of sensivity, specifity and accuracy from performance metrics?

Author Response

Dear Reviewer,

Thank you for reviewing our manuscript and for your valuable feedback. We appreciate your constructive comment and have addressed it in the attached response.

Given the minor nature of the comment, we have ensured that the necessary clarification has been incorporated into the manuscript. We hope this meets your expectations and are happy to provide further clarification if required.

Thank you again for your time and effort.

Best regards,

Martin Reinicke

Round 2

Reviewer 1 Report

Comments and Suggestions for Authors

The authors have satisfactorily addressed all the comments raised by reviewers and substantially improved the overall quality of the article. Therefore I recommend acceptance of this article for publication in IJMS.